# Seed Security Factors Driving Farmer Decisions on Uptake of Tissue Culture Banana Seed in Central Uganda

**Lucy Mulugo [1],\*, Florence Birungi Kyazze [1], Paul Kibwika [1], Bonaventure Aman Omondi [2] and Enoch Mutebi Kikulwe [3]**

[1]  Department of Extension and Innovation Studies, College of Agricultural and Environmental Sciences, Makerere University, P.O. Box 7062, Kampala, Uganda; fbirungikyazze@gmail.com (F.B.K.); pkibwika@caes.mak.ac.ug (P.K.)

[2]  Bioversity International C/O IITA/Benin Research Station, 08 Boite Postale, Cotonou 0932, Benin; b.a.omondi@cgiar.org

[3]  Bioversity International, Katalima Road, Naguru, P.O. Box 24384 Plot 106, Kampala, Uganda; e.kikulwe@cgiar.org

\*  Correspondence: mulugo@caes.mak.ac.ug; Tel.: +25-(670)-280-1136

**Abstract:** Despite the promotion of tissue culture (TC) banana to curb the spread of diseases, farmer use of such quality planting material remains low. This study utilizes the Double-Hurdle model on cross-sectional data of 174 banana farmers in Central Uganda to analyze the drivers for uptake of TC banana plant materials. Results show acceptability ($\beta = 0.74$; $p < 0.01$), adaptability ($\beta = 0.69$; $p < 0.01$) and availability for farmer use ($\beta = 1.04$; $p < 0.01$) along with social influence, farmer competences and socioeconomic factors positively influence farmer uptake of the TC banana plantlets. For uptake intensity, the main drivers include acceptability ($\beta = 0.39$; $p < 0.05$), accessibility ($\beta = 0.39$; $p < 0.01$) and farmer competences. This study demonstrates that seed security factors with farmer competencies, social influence and socioeconomic factors influence farmer decisions on uptake of TC technology for banana production. Findings emphasize the need for more involvement of extension services and research institutions in the education and promotion of TC plants in farming communities. We recommend that banana TC developers and promoters focus attention on banana varieties that are acceptable and adaptable to farmer environmental conditions.

**Keywords:** seed security; banana tissue culture planting material; uptake; banana farmers; central Uganda

## 1. Introduction

The per capita food output in Sub-Saharan Africa (SSA) has considerably declined, thus, contributing to increased food and income insecurity [1,2]. At the center of the debate on food and income, insecurity is the inability of the smallholder farmer to use quality plant material void of pests and diseases [3]. In the SSA region, the propagation system is characterized by both formal and informal plant material supplies. For banana (*Musa* spp.), the majority of smallholder farmers depend on the informal supply (including home-saved material from previous season harvests) [4]. For example, over 90% of banana farmers in East and Central Africa rely on suckers sourced from friends, neighbors, relatives and/or their own fields to either expand or establish new banana plantations [5,6]. The high prevalence of pests and disease in the home-saved plant material, however, has necessitated research and development of expert practitioners to increase the use of high-quality, formal supplies of plant material [7]. Such an approach, embedded in the farmers' social, cultural environment,

guarantees quality banana propagation material [8]. One such effort is the development of tissue culture (TC) planting materials (TC banana planting materials, TC banana plantlets and TC seed are used interchangeably throughout the manuscript), which are always free of pests and diseases.

The development of banana tissue culture is a response strategy for addressing the dual challenges of (1) supplying disease-free plant material and (2) enhancing farm-level yields in banana production. First, banana is a major staple for more than half of Uganda's population, and it provides a wide range of products (animal feeds, charcoal briskets, crafts, construction materials, etc.) which significantly contribute to food and income security of the populace and consequently to national development [9]. Despite the value and benefits derived from bananas, diseases such as banana *Xanthomonas* wilt (BXW) threaten its survival in the country. Between 2002 and 2005, BXW caused losses equivalent to 61.1 million US dollars to the country, mainly associated with the East African highland banana (EAHB) "Matooke" (AAA-EAHB genome) and the "Kayinja" beer banana (ABB genome) [10]. As such, farmer use of quality banana material (e.g., tissue culture plantlets) is considered a vital component for increasing banana survival and boosting agricultural productivity in the country.

Tissue culture (TC) generated banana plants are presumed to be free of BXW and recommended for the establishment of clean banana plantations [7]. However, the use of TC banana plantlets remains low [11], at less than 7% of the total banana production in the country [7]. For instance, a study by Akankwasa et al. [12] reveals that only two hundred and fifty mother gardens had been established in Uganda, and 40,000 tissue-cultured plantlets were distributed to banana farmers. Further, results indicate that merely 6% of banana farmers are willing to use TC banana planting material. Existing studies [13,14] have mainly focused on extrinsic (mainly socioeconomic) factors to explain the uptake of TC banana planting materials. However, studies that examine the role of security factors (availability, accessibility, acceptability and adaptability) in light of the farmer environment (competence, social influence and farmer's socioeconomic factors still contribute to the lack of TC plant use and necessitate further investigation.

Research is well established in root and tuber crops [15,16]. Studies on the farmer environment to explain farmer behavior in the uptake of agricultural technologies [17–19] have been conducted, but most of the existing research is qualitative. Research is needed to understand quantitative factors influencing uptake of agricultural technologies such as banana TC plantlets. Particularly, quantitative data are needed on how TC plant availability, accessibility, acceptability and adaptability combine with the farmer environment to influence the uptake of TC banana technology. The objective of this study is to explore the role of seed security factors and the farmer environment in the uptake of TC banana planting materials. Results will be important in designing interventions that ensure the sustainability of the banana crop in central Uganda.

## 2. Conceptual Framework, Study Area and Methods

### 2.1. Conceptual Framework

The uptake of agricultural technologies like TC banana planting materials is a complex, nonlinear process influenced by multiple factors. As such, the use of a single theory to analyze farmer decision-making in using tissue culture banana planting materials does not provide a holistic picture of the uptake process. Thus, we developed a conceptual framework (Figure 1) that encompasses seed security factors and the farmer environment to analyze the uptake of TC banana plantlets.

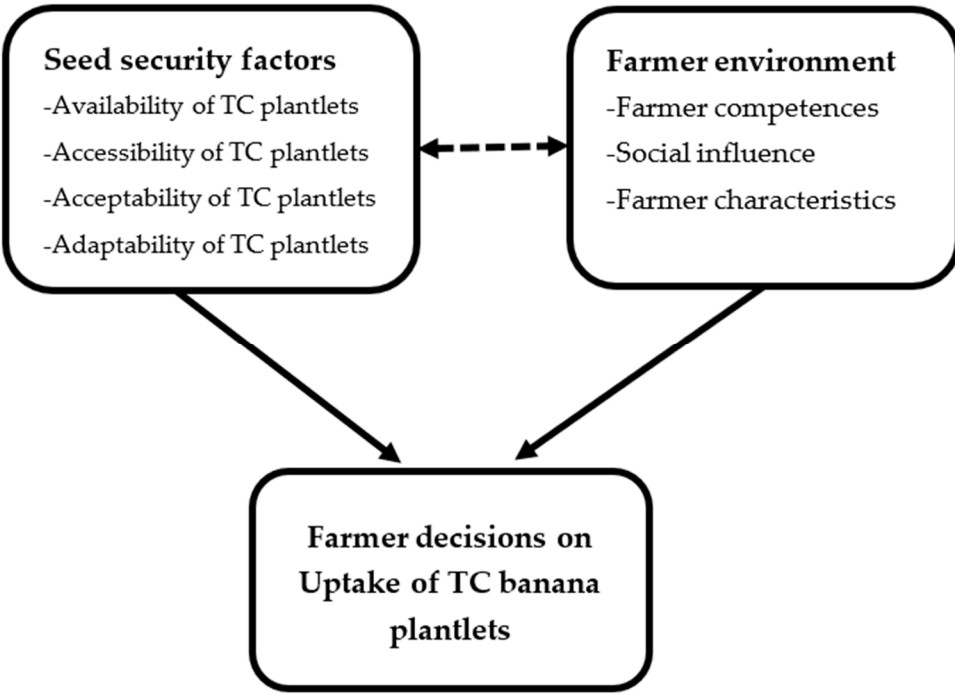

**Figure 1.** Seed security factors and farmer environment influencing uptake of TC banana plantlets (source: developed by authors).

The seed security framework with a focus on root tuber and banana (RTB) crops [20–22] explains that for sustained uptake of certified planting materials in a seed system (a seed system is the network of stakeholders involved in producing and planting seed (including vegetative planting material) of a particular crop in a certain area [15,20]), the factors:—availability, accessibility and quality are of relevance. It has been argued that the focus of seed security factors stimulates farmer confidence in the uptake of technologies and is of utmost relevance in identifying why uptake efforts fail (or succeed), thus aiding in the more effective design of future efforts [21]. Studies on seed security factors exist for potato [23,24], yam [25], cassava [26,27] banana and plantain [28,29] and sweet potato [30,31]. These studies reiterate the relevance of seed security factors in guiding programs that encourage the uptake of newly introduced technologies.

Availability indicates a sufficient and timely supply of propagation material from existing and functional sources to farmers [20]. Accessibility pertains to farmers' ease of acquiring TC banana planting material. This refers to whether farmers have the financial capital to purchase the plantlets and the feasibility of transporting the banana planting materials from the TC sources to destined localities [20]. Provision of information pertaining to seed technologies has also been found to be key in aiding farmers to access seed [15,23,24]. Acceptability relates to the provision of preferred and desirable seed varieties that are acceptable to meet farmers' tastes and preferences [22,32]. In accordance, Mulugo et al. [33] attest to banana farmers in central Uganda having desirable banana varieties preferred based on taste, aroma, color and texture. Similarly, Akankwasa et al. [12] found that banana taste, flavor, texture and color were key in determining consumers' likelihood of purchasing hybrid banana varieties in the four regions of Uganda.

Adaptability refers to the ability of TC banana plantlets to perform well in newly introduced environments and farming conditions. Kilwinger et al. [5] cite prolonged drought effects on banana productivity in central Uganda; Nyombi [34] mentions low soil fertility in the same region. However, previous studies tend to exclude TC established plantations, whose plantlets are characteristically fragile and sensitive to harsh environmental conditions [35]. Sinja et al. [36] confirm that farmers will only take up a technology that is best adapted to their environment. Thus, it is imperative that

TC planting materials adapt to the prevailing soil and environmental conditions in the study region. For this study, adaptability measures drought tolerance capabilities of TC established.

Theories on competence development posit knowledge, attitude and skill to be important factors that determine individual capabilities [37,38]. Competences consist of integrated pieces of knowledge, skills and attitudes that can be used to perform a task successfully, such as the uptake of TC plantlets [39]. Scholars [39,40] have shown skills to be interwoven with knowledge and viewed conjointly as doing or acting in practice. Meijer et al. [19] refer to knowledge as factual information and understanding of how a new agricultural technology works and what benefits the farmers can derive from it. Ugochukwu and Phillips [41], Kuehne et al. [42] and Meijer et al. [19] indicate that farmers' knowledge about the existence of a new technology extends to how to apply it and what the outcomes are in terms of products, yield, potential benefits, risks and costs. In this regard, knowledge is operationalized in this study to include elements of skill and technical knowledge that a farmer needs to grow TC plant materials, as well as the application of local technical knowledge to control BXW. In accordance with the tripartite theory, attitudes have three components- affective, behavioral and evaluative [43,44], which are key for the uptake of agricultural technologies. The affective domain refers to the emotional response of liking/disliking an object [45]. In this context, the TC plantlets. The behavioral component is a verbal or overt tendency by an individual [46] consisting of actions or observable responses that are the result of the object. The evaluative component, on the other hand, constitutes an individual's opinion of either belief or disbelief about a technology. For this study, attitude captures the affective and evaluative domains that include farmers' passion for growing TC plants and their perception of how the use of TC plants may reflect their personality to other people in their community.

Social influence refers to the degree to which a farmer perceives that relevant people believe that he or she should use agricultural technology [17]. Studies [47,48] attest to social influence triggering individuals' behavioral intentions to use new technologies. Thus, for this study, social influence captures the elements of farmer persuasion by members of farmer groups, faith-based leaders and community elders/leaders to grow bananas using TC planting material.

Previous research [12,49,50] shows the importance of farmer socioeconomic factors or characteristics (sex, age, education, farming experience, farm size, membership to farmer groups, and accessibility to agricultural extension services) on farmer uptake of agricultural technologies. Thus, these factors are incorporated into this study to evaluate whether they are critical in the uptake of TC banana plants. It is assumed that these factors in the presence of seed security factors could differently be influencing the uptake of TC banana seed.

## 2.2. Study Area

The study was conducted in Luweero and Mukono districts (Figure 2) in Central Uganda, where TC banana planting materials have been promoted for more than a decade. The two districts experience a high prevalence of BXW [51] despite numerous interventions to curb the disease. In each district, villages that hosted the community-based TC nurseries—Gonve and Nambi villages were selected on the assumption that proximity to TC nurseries enhances farmer physical access to TC banana planting materials. Farmers in the two villages were linked to TC laboratories (from whence they obtained TC plantlets) through farmer-managed community-based TC nurseries.



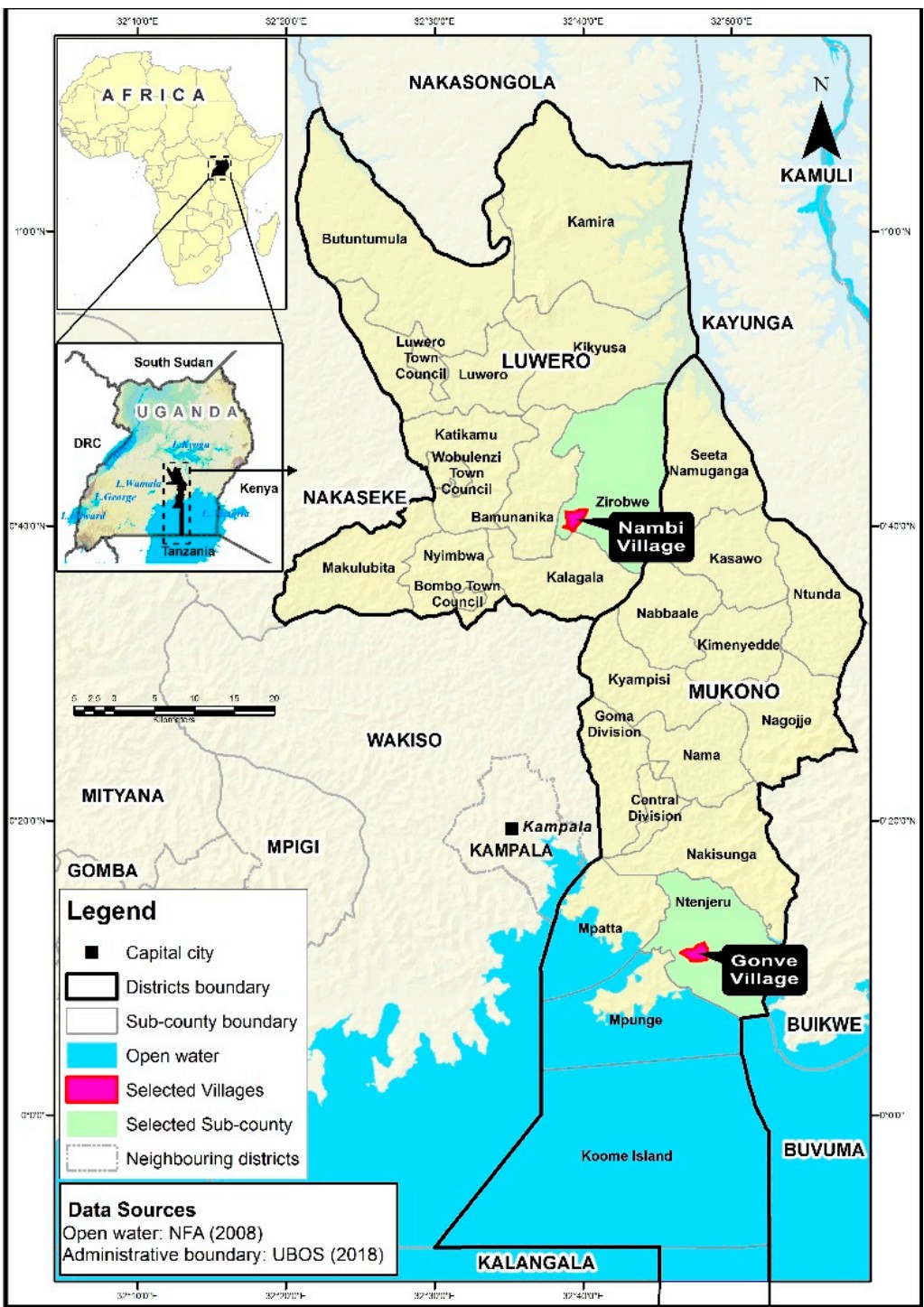

**Figure 2.** Map showing the study areas.

### 2.3. Study Design and Sample Selection

This study employed a quantitative research design. Data were obtained through a cross-sectional survey conducted in April and May 2018. The unit of analysis was the individual banana farmer. A questionnaire was administered to every respondent to generate data on variables of interest.

With the help of local council village chairpersons, names of all banana farmers from the selected villages that had been trained by The International Institute for Tropical Agriculture (IITA) between 2008 and 2011 (comprising adopters and potential adopters of the TC banana technology) were compiled to

generate the sampling frame with a total of 340 banana farmers (Table 1). Unlike Mulugo et al. [17], who sampled users and non-users of the TC banana plantlets, here we selected adopters and potential adopters of the TC planting materials. The farmers had been trained on how to grow and manage TC banana plantlets, business and marketing and the establishment of farmer cooperatives linked to the TC community nurseries. From a predetermined sample size of 174 banana farmers, estimated using the method suggested by Krejcie and Morgan [52], respondents were proportionately drawn from each of the study villages, as shown in Table 1 below.

**Table 1.** Number of farmers selected for the study.

| Location | Total Number of Farmers Listed | Number of Farmers Selected |
|---|---|---|
| Village: Nambi Luwero district | 120 | 68 |
| Village: Gonve Mukono district | 220 | 106 |
| Total | 340 | 174 |

## 2.4. Data Collection

Primary data were collected from the 174 randomly selected banana farmers using a semi-structured questionnaire. The questionnaire was pretested for reliability and suitability and then modified for clarity and sequencing of questions based on the pretest experiences and results. Data were collected through face-to-face interviews with the selected banana farmers. Data were collected on seed security factors, farmer competencies and farmer characteristics related to uptake of the TC banana planting materials.

Measures

Availability was measured based on statements (e.g., "TC is available in sufficient quantities"). Accessibility was measured based on 3 items (e.g., "The TC nursery operator provides information on how to plant TC plantlet"), acceptability was measured based on 3 items (e.g., "TC banana plantlets is of desirable banana varieties"), and adaptability was measured based on 2 items (e.g., "TC banana plantlets thrive in all soil types"). To measure farmer competencies, 8 statements were used. Knowledge was measured with 5 items (e.g., "I have sufficient technical knowledge to grow TC banana plants"), and attitude was measured with 3 items (e.g., "I have enough passion for growing TC banana plants"). The social influence variable had 3 items. A sample item reads as follows: "If I am informed about TC by a community leader, then I can use it as banana seed". Each item was measured on a five-point rating scale from 1 = least and 5 = highest.

In addition, the questionnaire included demographic characteristics of the farmers: These included age in years, the highest level of formal education in years, the gender of the farmer, farm size in acres, experience in banana farming in years, access to agricultural extension services, membership to farmer groups, receipt of agricultural information, land apportioned for crop production and banana cultivation.

## 2.5. Analytical Framework

Preceding the regression analysis, principal component analysis (PCA) was carried out for data reduction and extraction of variables. Based on the criterion of eigenvalues being greater than 1 [53] in the PCA, underlying dimensions among the farmers' perceptions of the TC plantlets' attributes were generated.

This study breaks the banana farmer decision-making into two stages. In the first stage, the banana farmer's decision is either uptake of the banana TC technology or not. In the second stage, the farmer's decision is on how much of the TC plant material to use (uptake intensity), which in this study is

represented by the size of land allocated to the banana TC plantlets. Data were analyzed using Cragg's double-hurdle model [54] consisting of a two-stage regression in which the first stage is a probit model to analyze factors influencing a binary decision on uptake of technology and the second stage is a truncated model which analyzes factors that affect uptake intensity. Farmer uptake of banana TC plantlets was analyzed in the following equation:

$$Y = \beta_0 + \beta_i[Seed\_Sec]_i + \beta_j[Farm\_Envt]_j + \varepsilon \tag{1}$$

where $Y$ represents farmer's decision-making for the uptake of banana TC plantlets uptake = 1; no uptake = 0) and in the second stage, $Y$ equals the uptake intensity. $[Seed\_Sec]$ is a vector of plant security factors, which includes perceived acceptability, accessibility, adaptability and availability of the plantlets. $[Farm\_Envt]$ comprises three components: (1) social influence, (2) farmer competencies, and (3) farmer socioeconomic factors or characteristics. $\beta_0$ is the constant while $\beta_i$ represent the various coefficients of the security factors ranging 1–4. $\beta_j$ are the coefficients of the farmer environment factors ranging from 1 to 9 and $\varepsilon$ is the error term. The a priori hypothesized signs of the coefficients are shown in Table 2.

**Table 2.** Apriori signs of explanatory variables used in the study.

| Variable | Apriori Sign | Reference |
|---|---|---|
| Availability of TC banana plantlets | (+) | [15,16] |
| Accessibility of TC banana plantlets | (+) | [15,16] |
| Acceptability of TC banana plantlets | (+) | [22,32] |
| Adaptability of TC banana plantlets | (+) | [22,32] |
| Farmer Knowledge on TC plantlets | (+) | [41,42] |
| Social Influence | (+) | [17,47,48] |
| Farmer Attitude towards TC plantlets | (+) | [19,43] |
| Gender | (+) | [12,50] |
| Age | (+/−) | [12] |
| Education | (+) | [12,49] |
| Farm size | (+) | [12,50] |
| Land allocated to other crops | (-) | [12] |
| Group membership | (+) | [50] |
| Extension services | (+) | [12,50] |

## 3. Results and Discussion

Table 3 presents a summary of key descriptive statistics of study respondents. Results show an equal number of male and female farmers were interviewed. About 33% of respondents have adopted the TC banana planting materials. The average age of banana farmers in the study communities was 43 years, with an average of eight (8) years of formal education. The average farm size for all farmers was 4.3 acres, with an average of about 2.6 acres allocated to crops and about 1.1 acres to banana production.

**Table 3.** Description and summary statistics of respondent characteristics.

| Variable | Mean (SD) | Min | Max | Percentage (%) |
|---|---|---|---|---|
| Sex (1 = male, 0 = female) | 0.53 (0.50) | 0 | 1 | - |
| Age of farmer (years) | 42.83 (13.63) | 18 | 93 | - |
| Formal education (years) | 7.74 (3.45) | 1 | 16 | - |
| Banana farming experience (Years) | 18.10 (13.10) | 0 | 70 | - |
| Farm size (acres) | 4.27 (5.75) | 0.50 | 69 | - |
| Land allocated to crops (acres) | 2.60 (2.41) | 0.42 | 22.75 | - |
| Land under banana cultivation (acres) | 1.05 (1.08) | 0.13 | 10 | - |
| Availability of banana TC plantlets | 3.07 (1.166) | 1 | 5 | - |
| Accessibility of banana TC plantlets | 2.94 (1.027) | 1 | 5 | - |
| Acceptability of TC banana plantlets | 3.60 (0.946) | 1 | 5 | - |
| Adaptability of TC banana plantlets | 2.74 (0.971) | 1 | 5 | - |
| Farmer knowledge about TC plantlets | 2.45 (1.365) | 1 | 5 | - |
| Farmer attitude towards TC plantlets | 3.13 (1.211) | 1 | 5 | - |
| Social influence | 3.83 (0.845) | 1 | 5 | - |
| Farmer attitude towards TC plantlets | 3.13 (1.211) | 1 | 5 | - |
| Banana TC users (Yes = 1) | - | - | - | 33.3 |
| Access to extension services (Yes = 1) | - | - | - | 38.5 |
| Receipt of agricultural information (Yes = 1) | - | - | - | 15.5 |
| Membership to groups (Yes = 1) | - | - | - | 39.7 |

Standard deviation (SD) is in parenthesis.

Results of the Kaiser–Meyer–Olkin (KMO) measure (0.815) and Bartlett's test of sphericity ($\chi$ = 3804.143; $p$ < 0.001) (Table 4) indicate sampling adequacy and suitability of the data for factor analysis [55]. Principal component results show that seven (7) extracted factors explained 76.2% of the total variance in the principal components. Specifically, as presented in Table 4, the variance extracted ranged from 32.4% (for farmer knowledge) to 4.4% (TC plant adaptability). In addition, the factor loadings for the extracted variables ranged from 0.508 to 0.925, and thus, convergent validity was confirmed [56]. Lastly, the Cronbach's alpha values (Table 4) range from 0.600 to 0.992, signifying adequacy of internal consistency, and thus, confirmation of measurement validity [57] in this study.

**Table 4.** Loadings of perception and attitude factors for uptake of TC banana planting materials (*n* = 174).

| Item Description | Cronbach's Alpha | Factor Loadings * | | | | | | |
|---|---|---|---|---|---|---|---|---|
| | | Factor 1 Knowledge | Factor 2 Availability | Factor 3 Social Influence | Factor 4 Attitude | Factor 5 Accessibility | Factor 6 Acceptability | Factor 7 Adaptability |
| Apply organic fertilizer to TC plantlets | 0.992 | 0.925 | | | | | | |
| Apply organic pesticide to TC plantlets | | 0.921 | | | | | | |
| Use LTK in BXW management with TC plantlets | | 0.883 | | | | | | |
| Sufficiency in technical knowledge to grow TC plantlets | | 0.538 | | | | | | |
| Skilled to grow TC plantlets | | 0.601 | | | | | | |
| Existence of TC nurseries/sources | 0.915 | | 0.883 | | | | | |
| Availability of functional TC nurseries/sources | | | 0.881 | | | | | |
| Availability of TC plantlets in sufficient quantities | | | 0.790 | | | | | |
| Availability of TC plantlets in time | | | 0.710 | | | | | |
| If informed about TC plantlets by a community leader, then I can use it as planting material | 0.939 | | | 0.912 | | | | |
| If informed about TC plantlets by a faith-based leader, then I can use it | | | | 0.906 | | | | |
| If informed about TC plantlets by a member of a farmer group, then I can use it as planting material | | | | 0.844 | | | | |
| According to people that are important to me, I should use TC plantlets | 0.834 | | | | 0.618 | | | |
| I have enough passion for planting TC plantlets | | | | | 0.658 | | | |

**Table 4.** *Cont.*

| Item Description | Cronbach's Alpha | Factor Loadings * | | | | | | |
| --- | --- | --- | --- | --- | --- | --- | --- | --- |
| | | Factor 1 Knowledge | Factor 2 Availability | Factor 3 Social Influence | Factor 4 Attitude | Factor 5 Accessibility | Factor 6 Acceptability | Factor 7 Adaptability |
| Use of TC plantlets reflects my personality to other farmers | | | | | 0.751 | | | |
| Access to information on how to grow TC plantlets | 0.839 | | | | | 0.851 | | |
| Access to information on how to manage TC plantlets | | | | | | 0.864 | | |
| Affordability of TC plantlets | | | | | | 0.666 | | |
| Farmer desirability of banana varieties supplied as TC plantlets | 0.651 | | | | | | 0.786 | |
| Appropriateness of size of TC plantlets | | | | | | | 0.664 | |
| Acceptability of taste of food from TC plantlets | | | | | | | 0.681 | |
| Drought tolerance capability of TC plantlets | 0.600 | | | | | | | 0.725 |
| Capability of TC established plantations to last long | | | | | | | | 0.744 |
| Ability of TC plantlets to thrive in all soil types | | | | | | | | 0.508 |
| Eigenvalues | | 8.098 | 3.080 | 2.448 | 1.660 | 1.483 | 1.886 | 1.091 |
| % of variance explained | | 32.393 | 12.321 | 9.793 | 6.639 | 5.934 | 4.743 | 4.363 |

Kaiser–Meyer–Olkin measure of sampling adequacy = 0.815; approx. chi-squared = 3804.143. Bartlett's sphericity test: df = 300; $p < 0.001$.

### 3.1. Factors Associated with the Uptake Decision for Farmer Use of TC Banana Plantlets

Results of the first stage analysis (probit regression) show that farmer perceived acceptability ($\beta$ = 0.74; $p$ < 0.01) has a positive and significant influence on farmer decisions for the uptake of the banana TC planting material (Table 5). Similarly, perceived adaptability ($\beta$ = 0.69; $p$ < 0.01) and perceived availability ($\beta$ = 1.04; $p$ < 0.01) had a positive and significant influence on uptake. These findings are in agreement with other studies regarding the influence of security factors on the uptake of improved plant materials [15,20,21]. In terms of marginal effects, TC plant acceptability (0.062) implies that, on average, a 1% increment in farmer acceptance of banana TC plants increases the probability of farmer uptake by 6.2%. This is likely because farmers tend to prefer introduced varieties that are comparable to their local varieties with regard to desirable attributes. Similar results are also reported by Mulugo et al. [33] and Akankwasa et al. [12] for the case of Uganda.

**Table 5.** Factors that influence farmers' decisions regarding adoption of tissue culture banana planting materials: results of the probit model.

| Variable | Coefficient | Robust Std. Error | $p > z$ | Average Marginal Effects (dy/dx) |
|---|---|---|---|---|
| **Seed security factors** | | | | |
| Perceived acceptability of TC plantlets | 0.737 | 0.276 | 0.008 ** | 0.062 |
| Perceived accessibility of TC plantlets | 0.302 | 0.229 | 0.187 | 0.025 |
| Perceived adaptability of TC plantlets | 0.688 | 0.263 | 0.009 ** | 0.058 |
| Perceived availability of TC plantlets | 1.037 | 0.340 | 0.002 ** | 0.087 |
| **Farmer competences** | | | | |
| Knowledge about TC plantlets | 2.155 | 0.434 | <0.00 *** | 0.180 |
| Attitude to TC plantlets | 1.000 | 0.337 | 0.003 ** | 0.084 |
| **Farmer socioeconomic factors** | | | | |
| Social influence | 0.810 | 0.372 | 0.030 * | 0.068 |
| Sex of farmer | −1.377 | 0.562 | 0.014 * | −0.115 |
| Age of farmer | −0.088 | 0.297 | 0.767 | −0.007 |
| Education of farmer | 0.059 | 0.062 | 0.347 | 0.005 |
| Group membership | −0.496 | 0.551 | 0.369 | −0.042 |
| Access to extension services | −0.344 | 0.525 | 0.513 | −0.029 |
| Land allocated to crops | 0.341 | 0.154 | 0.027 * | 0.029 |
| Experience in banana farming | −0.013 | 0.025 | 0.624 | −0.001 |
| Constant | −1.016 | 1.158 | 0.381 | |
| Number of observations | 174 | | | |
| Log-likelihood ratio | −25.98 | | | |
| Wald $\chi^2$ | 169.56 *** | | | |
| Mean VIF | 1.8 | | | |

\* $p$ < 0.05, \*\* $p$ < 0.01, \*\*\* $p$ < 0.001; dependent variable: uptake measured as a binary.

For farmer perceived adaptability of TC plantlets (marginal effect = 0.058), an additional increase in adaptability is associated with a probability of a 5.8% increase in uptake. Notably, the importance of TC plant adaptability to drought and poor soil fertility is emphasized [5,34]. Lastly, the average marginal effect of TC plant availability (0.087) implies that an additional TC nursery is associated with a likelihood of an 8.7% increase in farmer uptake of TC plantlets. Related studies like Okechukwu and Kumar [27] on availing disease-resistant varieties in Africa and Kromann et al. [23] on the provision of quality seed in Ecuador have reported sufficient and timely availability of quality plant material to increase farmer uptake of plant technologies.

Farmer knowledge about TC plant cultivation ($\beta$ = 2.16; $p$ < 0.001) has a positive and significant effect on uptake. Similarly, farmer attitude ($\beta$ = 1.10; $p$ < 0.01) posits a positive and significant effect on uptake. These results concur with the theory that farmer competencies predict uptake [19,41,42].

In terms of marginal effects, the result on farmer knowledge about TC plants (0.180) implies that, on average, an increment in farmer skill and technical knowledge on how to grow TC plantlets increases the likelihood of farmer uptake by 18%. This finding validates the study by Atieno and Schulte-Geldermann [24] on public–private partnerships in the multiplication of plant materials in Kenya. Specifically, knowledge sharing and training by extension agents increase farmer uptake of new technology.

Similarly, our findings substantiate previous research about farmer attitudes towards the uptake of introduced agricultural technologies [58,59]. Particularly, Mekoya et al. [58] found Ethiopian farmers' positive attitude towards multipurpose fodder trees (for their feed value and contribution to soil conservation) enhanced farmer uptake of the agroforestry technology.

Social influence ($\beta = 0.81$; $p < 0.05$) also had a positive and significant influence on farmer decisions for uptake of the TC banana plants [19,60]. The average marginal effect (0.068) infers that having an influential person inform farmers about the importance of banana TC plants increases the likelihood of farmer uptake by 6.8%, suggesting that locally institutionalized mechanisms need to be promoted. Similar results are also reported by Mulugo et al. [17] and Wauters et al. [61] for Uganda and Belgium, respectively. Specific for Uganda, using faith-based leaders, political and community leaders urge farmers to use TC banana plantlets is a significant predictor of farmer intentions to use TC planting materials.

Contrary to most studies [12,62,63], the model estimates show that the gender of the farmer ($\beta = -1.38$; $p < 0.05$) has a significant effect on the uptake decision, indicating that women were more likely to use TC banana planting materials compared to men. A possible explanation for women's interest in TC banana planting material could be attributed to the distinct role that women play in ensuring food security for their families. Food security in central Uganda is culturally viewed as a women's role, and women must ensure that their homesteads are food secure (Sanya et al. [64]). The study finding is in line with Sanya et al. [64], who found female farmers to be 12.4% more likely to adopt hybrid banana varieties compared to male farmers. Nevertheless, with recent developments by the government being skewed to banana-based processing enterprises, it is likely that men will increasingly grow bananas to boost their sources of income.

The amount of land allocated to crop production ($\beta = 0.34$; $p < 0.05$) has a positive influence on the likelihood of farmer uptake of the TC banana technology. Based on the average marginal effect (0.029), increasing the land size by one acre increases the likelihood that farmers would try the TC planting materials by 2.9%. A possible explanation for this finding could be that farmers with larger farms may be more willing to take risks and devote portions of their land to growing bananas using TC plantlets, compared with those with smaller land areas. This is in line with most adoption studies [50,65,66] that found that farmers with larger farm sizes have more land to allocate to new agricultural technologies.

*3.2. Factors Associated with Intensity of Farmer Uptake of TC Banana Plantlets*

Results of second stage analysis (truncated regression) show that farmer perceived acceptability ($\beta = 0.39$; $p < 0.05$) and accessibility ($\beta = 0.39$; $p < 0.01$) of TC plants posit a positive and significant influence on farmer decisions (Table 6). Essentially, the result specifies that if farmers perceive that the varieties being promoted through TC match their preferred food attributes, then they are more likely to expand their banana plantations using such a technology. Previous studies [12,33] attest to this finding.

Similarly, farmer accessibility (marginal effect = 0.058) indicates that farmers with fairly priced plant material coupled with information on how to grow it promotes plantation expansion by 5.8%. This finding is in tandem with studies [15,23,24] confirming accessibility in terms of affordability and awareness creation to be crucial for use and extent of uptake of introduced plant technologies. Specifically, access to quality potato planting material in Kenya increased farmer uptake of the planting materials by 30–40%, leading to increased yields (5184 t/year of potatoes) and profits ($777,600) [24].

**Table 6.** Factors that influence the intensity of uptake of the TC banana plantlets: results of the truncated regression model.

| Variable | Coefficient | Robust Std. Error | $p > z$ | Average Marginal Effects (dy/dx) |
|---|---|---|---|---|
| **Seed security factors** | | | | |
| Perceived acceptability of TC plantlets | 0.390 | 0.179 | 0.029 * | 0.018 |
| Perceived accessibility of TC plantlets | 0.392 | 0.119 | <0.001 ** | 0.058 |
| Perceived adaptability of TC plantlets | 0.019 | 0.084 | 0.822 | 0.003 |
| Perceived availability of TC plantlets | 0.049 | 0.086 | 0.566 | 0.007 |
| **Farmer competences** | | | | |
| Knowledge about TC plantlets | 1.461 | 0.197 | <0.00 *** | 0.067 |
| Attitude to TC plantlets | 0.524 | 0.185 | 0.005 ** | 0.024 |
| **Farmer socioeconomic factors** | | | | |
| Social influence | −0.189 | 0.098 | 0.055 * | −0.028 |
| Sex of farmer | −0.874 | 0.347 | 0.012 * | −0.040 |
| Age of farmer | −0.001 | 0.058 | 0.927 | −0.001 |
| Education of farmer | 0.010 | 0.020 | 0.623 | 0.002 |
| Group membership | −0.119 | 0.175 | 0.497 | −0.018 |
| Access to extension services | 0.166 | 0.440 | 0.706 | 0.008 |
| Land allocated to crops | 0.218 | 0.031 | <0.00 *** | 0.032 |
| Farm size | 0.119 | 0.058 | 0.041 * | 0.006 |
| Experience in banana farming | 0.003 | 0.008 | 0.678 | 0.678 |
| Constant | −0.814 | 0.365 | 0.026 | |
| Number of observations | 58 | | | |
| Log-likelihood ratio | −34.39 | | | |
| Wald chi-squared | 187.44 *** | | | |
| Mean VIF | 1.88 | | | |

*$p < 0.05$, **$p < 0.01$, ***$p < 0.001$; dependent variable: proportion of land planted with banana TC seed.

Farmer knowledge about TC plant materials (β = 1.46; $p < 0.001$) and farmer attitude towards TC technology (β = 0.52; $p < 0.01$) both postulate a positive and significant influence on farmer decisions to allocate more of their land to TC banana production. The findings concur with previous research about farmer knowledge on introduced agricultural technologies to influence uptake and extent of use of improved crop varieties [19,24,41,42]. Similarly, farmer attitude result resonates with findings by [67] in Kenya, showing farmer attitudes to be key indicators in predicting uptake of aquaculture technologies among smallholder fish farmers.

Surprisingly, social influence (β = −0.19; $p < 0.05$) had a negative influence on farmer decisions to apportion more of their land to TC banana plantlets. This varied from the results of earlier research [47,68], which reported positive effects of social influence on farmer uptake of agricultural technologies. This probably is attributable to the cost implication of expanding banana cultivation using TC plant materials, inevitably requiring farmers to purchase plantlets, inputs and follow through with the recommended cultivation instructions that often are labor-intensive [14]. As such, it may be important that a related study on social influence is designed in a different context to confirm the actual impact of social influence on farmer intensity of use of agricultural technologies.

Although not a significant predictor of uptake, farm size (β = 0.12; $p < 0.05$) had a positive influence on uptake intensity of TC banana planting materials.

## 4. Conclusions

This study shows that farmers' decisions for the uptake of banana TC plants are positively and significantly influenced by seed security factors. From a practical perspective, the study contributes to results that show the importance of developers in the seed system in focusing on farmer desired

crop attributes. Further, the study emphasizes the need for more involvement of extension services and research institutions in education about cultivation and promotion of TC planting materials in the banana farming communities. This involvement could incorporate the use of community role models since social influence plays a pivotal role in increasing uptake. We recommend seed security factors (acceptability, accessibility, adaptability and availability), social influence and farmer competence (knowledge, skill and attitude) as variables to be considered in programs aimed at increasing farmer uptake for seed system technologies.

**Author Contributions:** L.M., F.B.K., P.K., B.A.O. and E.M.K. conceived and designed the study. L.M., F.B.K. and P.K. developed the data collection tools. L.M., F.B.K. and P.K. participated in field research. L.M. and F.B.K. analyzed and interpreted the data, E.M.K., and L.M. acquired funding. All authors participated in writing, reviewing and editing the article. All authors have read and agreed to the published version of the manuscript.

**Funding:** This research was funded by the CGIAR Research Program (CRP) Roots Tubers and Bananas (RTB) through the "RTB Seed Systems Project" grant number 96202-500-A1050-B100461-C100078-UGAND and the Regional Universities Forum for Capacity Building in Africa (RUFORUM) with funding from the Carnegie Corporation of New York grant number RU/2016/Carnegie/DRG/009.

**Acknowledgments:** We are grateful to the banana farmers in the study communities who participated in this study. Our appreciation goes to Emmanuel Ngolobe, Eva Tereka, Ruth Nakintu, Ssebunya Mohammed Ali and Ssenyonjo Michael for their participation in data collection. Justine Kyobe, Sserunkuma Margaret, Nakafu Margaret Kyasa (Mukono district) and Kiggundu (Luweero district) are appreciated for their role in mobilizing the respondents. This research was undertaken as part of the CGIAR Research Program on Roots, Tubers and Bananas (RTB).

**Conflicts of Interest:** The authors declare no conflict of interest.

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
