# Peer review of "Seed Security Factors Driving Farmer Decisions on Uptake of Tissue Culture Banana Seed in Central Uganda"

_sustainability, doi:10.3390/su122310223_

Round 1

Reviewer 1 Report

This is a good paper. My main concern is the use of the word "seed" throughout the paper when, in fact, tissue culture does not involve any seed. This needs to be corrected. Also, the paper is redundant and/or verbose in many sections. I've made numerous editing suggestions and some comments in the attached PDF which should improve the paper quite a bit.

Author Response

Response to Reviewer 1 Comments

Point 1: The use of the word "seed" is very misleading since this paper is about vegetative propagation and seed is not used at all. This term needs to be replaced throughout with "plant material" or "propagation material" or "vegetative material"

Response 1: The word “seed” has been replaced with TC banana planting materials and TC banana plantlets throughout the manuscript, with a few exceptions. Clarification is given in foot note two (2) where the term “seed system” is defined according to references [15, 20] as; A seed system is the network of stakeholders involved in producing and planting “seed”  (including vegetative planting material) of a particular crop in a certain area.

Clarification is also given under lines 103 to 110.

Point 2: Suckers and seed are very different types of plant material. In this paper, I think the use of the term "seed" should be changed to "plant material" because it's clear that SEED is not being used at all.

Response 2: Footnote two (2) [according to references 15 and 20] as explained above provide clarification.

Point 3: What is meant by farmer characteristics? personality? resources? attitude?

Response 3: Farmer characteristics have been changed to farmer’s socio-economic factors throughout the manuscript.

Point 4: Again, the word "seed" is very misleading since banana are rarely propagated by seed. Tissue culture uses NO seed whatsoever. The word "seed" must be removed from this paper unless you add a sentence in the introduction to explain the difficulties/impossibilities of propagating banana species from seed.

Response 4: This has been clarified as explained under points one (1) and two (2) above.

Point 5: Therefore, it is not possible to fully assess plant availability/accessibility since these factors would not be challenging in this location.

Response 5: Plant availability/accessibility were assessed at multi dimensions. Specifically, availability was assessed using 4 dimensions (i.e. existence of TC nurseries, function-ability of the nurseries, availability of TC plantlets in sufficient quantities and on time). Secondly, accessibility was assessed at 3 levels (i.e. farmer affordability of the TC plantlets and access to information on how to plant and grow the TC banana planting materials.

Point 6: Instead of this paragraph, a table with the list of statements would be more informative.

Response 6: A list of statements used for each construct is summarized in Table 4.  We have not included another table because Sustainability restricts the number of Tables in a manuscript to six (6).

Point 7: This belongs in the Results section.

Response 7: As recommended, the table describing the summary statistics of respondents has been shifted to the results section.

Point 8: This paper would be much improved by reducing the repetitive text (just highlight the most significant findings and the tables can tell the rest), and add more details from the citations - for example, what did they study? how much of an increase in uptake did they find

Response 8: The paper has been improved by reducing the repetitive text and highlighting the most significant findings. Details from citations have also been provided whilst discussing results. These details entail what the studies were about and also on how much of an increase in uptake scholars found (reference lines: 341 to 352; 366 to 372) in the revised version of the manuscript.

Point 9: no need to repeat the results - this section should focus on implications for future banana production and need for future research (if any).

Response 9: Results in the conclusion section of the manuscript have been deleted. The section has been re-written, solely focusing on implications for future banana production and need for future research.

Reviewer 2 Report

General comments:

English language professional editing highly needed.

Clearly identify the aim of the study as well as objectives, and then each objective should be discussed in the Discussion part.

Where the 174 farmers the same as 248 farmers from the study Mulugo, Lucy, Florence Birungi Kyazze, Paul Kibwika, Enoch Kikulwe, Aman Bonaventure Omondi, and Susan Ajambo. "Unravelling technology-acceptance factors influencing farmer use of banana tissue culture planting materials in Central Uganda." African Journal of Science, Technology, Innovation and Development 12, no. 4 (2020): 453-465.?

If yes please explain in the methods part. Also please, in case the farmers are the same as in the previous study explain the differences from the previous study.

Was Figure 1 developed by the authors or is adapted from another study?

Please explain more why gender has a significant influence on using TC banana plantlets?  

In the discussion part please elaborate more on why your study is different from the previous and what results its shows. Also please highlight how your study can help sustainable agriculture to be implemented and what role does it have in the sustainable growth of bananas. recommendations for decision-makers based on your results could be highlighted in the conclusion part.

Small suggestions:

Line 39 where is the close of the brackets?

Line 37 Explain better? Rephrase and explain “For example, over 90% of banana farmers in East and Central Africa rely on suckers”

Author Response

Response to Reviewer 2 Comments

Point 1: English language professional editing highly needed.

Response 1: The services of a professional English language editor have been utilised in editing a revised version of the manuscript.

Point 2: Clearly identify the aim of the study as well as objectives, and then each objective should be discussed in the Discussion part.

Response 2: The aim and objective of the study have been clarified; (lines 68 to 72) of the revised version of the manuscript attest to this.

Point 3: Were the 174 farmers the same as 248 farmers from the study Mulugo, Lucy, Florence Birungi Kyazze, Paul Kibwika, Enoch Kikulwe, Aman Bonaventure Omondi, and Susan Ajambo. "Unravelling technology-acceptance factors influencing farmer use of banana tissue culture planting materials in Central Uganda." African Journal of Science, Technology, Innovation and Development 12, no. 4 (2020): 453-465.? If yes please explain in the methods part. Also please, in case the farmers are the same as in the previous study explain the differences from the previous study.

Response 3: Clarity is provided under lines 217 to 224 of the manuscript.

 Point 4: Was Figure 1 developed by the authors or is it adapted from another study?

Response 4: Figure 1 is developed by the authors. This is specified; (line 101 and 102).

Point 5: Please explain more why gender has a significant influence on using TC banana plantlets?  

Response 5: A detailed explanation is provided under lines 341 to 350 in the manuscript.

Point 6: In the discussion part please elaborate more on why your study is different from the previous and what results its shows. Also please highlight how your study can help sustainable agriculture to be implemented and what role does it have in the sustainable growth of bananas. Recommendations for decision-makers based on your results could be highlighted in the conclusion part.

Response 6: The discussion part highlights differences between the previous study and the current one. For example, it points out that the previous study assessed factors influencing farmer intentions to use the TC planting materials whilst the current focuses on farmer use and uptake intensity of TC plantlets in the context of seed security factors and the farmer environment. We have also highlighted how our study can aid in the sustained growth of bananas and provided recommendations for decision-makers (lines 398 to 407).

Point 7: Line 39 where is the close of the brackets?

Response 7: A close bracket has been inserted.

Point 8: Line 37 Explain better? Rephrase and explain “For example, over 90% of banana farmers in East and Central Africa rely on suckers”

Response 7: Lines 37 to 43 have been rephrased to read: “For example, over 90% of banana farmers in East and Central Africa rely on suckers sourced from friends, neighbors, relatives and/or their own fields to either expand or establish new banana plantations [5, 6]. The high prevalence of pests and disease in the home-saved plant material, however, has necessitated research and development of expert practitioners to increase the use of high-quality, formal supplies of plant material [7]. Such an approach, embedded in the farmers’ social cultural environment, guarantees quality banana propagation material [8]”.

Round 2

Reviewer 2 Report

Dear authors, please improve the English language of the manuscript. I still found mistakes and it is inappropriate to publish in a high-ranking journal with English language mistakes. 

Line 103 "with foci" should be "with focus"

Line 108 "exist in potato" should be "exist for potato"

The abbreviation local technical knowledge (LTK) is only once used, so better not to do an abbreviation.

Availability and Accessibility one time you write Italic and the other time not, make it consistent throughout the text.

Line 246 "scale with" should be "scale from " 

Author Response

Response to Reviewer's comments:

Point 1: Line 103 "with foci" should be "with focus"

Response 1: Line 103 "with foci" has been changed to " with focus"

Point 2: Line 108 "exist in potato" should be "exist for potato"

Response 2: Line 108 "exist in potato" has been changed to "exist for potato"

Point 3: The abbreviation local technical knowledge (LTK) is only once used, so better not to do an abbreviation.

Response 3: The abbreviation local technical knowledge (LTK) has been deleted.

Point 4: Availability and Accessibility one time you write Italic and the other time not, make it consistent throughout the text.

Response 4: Availability and accessibility are used consistently throughout the text. The italics has been removed.

Point 5: Line 246 "scale with" should be "scale from " 

Response 5: Line 246 "scale with" has been changed to "scale from".